# Ancient Mitogenomes Reveal Stable Genetic Continuity of the Holocene Serows

**DOI:** 10.3390/genes14061187

**Published:** 2023-05-29

**Authors:** Shiwen Song, Bo Xiao, Jiaming Hu, Haifeng Lin, Zhicheng Du, Kunpeng Xiang, Dong Pan, Xindong Hou, Junxia Yuan, Xulong Lai, Guilian Sheng

**Affiliations:** 1School of Environmental Studies, China University of Geosciences, Wuhan 430078, China; 1202010362@cug.edu.cn (S.S.);; 2State Key Laboratory of Biogeology and Environmental Geology, China University of Geosciences, Wuhan 430078, China; xiaobo@cug.edu.cn (B.X.);; 3School of Earth Science, China University of Geosciences, Wuhan 430074, China; 4Guizhou Institute of Geological Survey, Guiyang 550081, China; 5Palaeontological Fossil Conservation Center, Qinggang County, Suihua 151600, China; 6Faculty of Materials Science and Chemistry, China University of Geosciences, Wuhan 430078, China

**Keywords:** serow, ancient DNA, genetic continuity, divergence, population dynamics

## Abstract

As one of the remaining species of Caprinae only found in Asia, serows (*Capricornis*) and their classification and conservation have received increasing attention in recent years. However, their evolutionary history and population dynamics are not yet clear. To shed light on these topics, we report the first near-complete ancient mitochondrial genomes from two serow sub-fossils (CADG839 and CADG946) dating to 8860 ± 30 years and 2450 ± 30 years, and incorporate the newly obtained mitogenomes into the dataset of living serows (18 complete mitochondrial genomes drawn from National Center for Biotechnology Information, NCBI) to investigate their relationships and evolution. Phylogenetic results support four clades of serows that can be further divided into five subclades, indicating higher genetic diversity than previously thought. Notably, our two ancient samples do not form a separate branch but belong to *Capricornis sumatraensis* clade A together with modern individuals, which suggests genetic continuity between ancient and modern serows. Furthermore, our results suggest that the maternal divergences of serows occurred at the beginning of the Pleistocene. Bayesian estimation indicates that the first divergence among all serows happened approximately 2.37 Ma (95% highest posterior density, HPD: 2.74–2.02 Ma) when Japanese serow (*Capricornis crispus*) appeared, while the last divergence occurred within the Sumatran serow (*C. sumatraensis* clade A and B) around 0.37–0.25 Ma. Additionally, we found the effective maternal population size of *C. sumatraensis* increased around 225–160 and 90–50 ka, then remained stable since 50 ka. Overall, our study provides new insights into serow phylogeny and evolutionary history.

## 1. Introduction

Serows belongs to the genus *Capricornis* Ogilby, 1837, which is within the Caprinae Gray, 1821, family and the Bovidae Gray, 1821, family. This genus is typically represented by tropical and subtropical species. It possibly originated in Central Asia in the mid to late Pliocene (about 3.3–2.5 Ma). The extant serows mostly live in China, Japan and Southeast Asia [1,2,3,4,5]. As the most widely distributed member of serows in China and Southeast Asia, the Sumatran serow (*C. sumatraensis*) once reached Northern China (39° N), while the northernmost group of extant mainland serow lives at 30° N in China [1,3]. Unfortunately, population isolation due to habitat fragmentation and human hunting has made their population size reduce quickly, and it is listed as a vulnerable (VU A2; C1) species on the International Union for the Conservation of Nature Red List [1].

Based on morphological characteristics, serows were divided into seven species: *C. crispus*, *Capricornis swinhoei*, *Capricornis rubidus*, *C. sumatraensis*, *Capricornis milnedwardsii*, *Capricornis thar*, and *Capricornis maritimus* [6,7,8]. However, recent investigation based on molecular phylogeny suggested that only four species exist in *Capricornis*, i.e., *C. crispus*, *C. swinhoei*, *C. rubidus*, and *C. sumatraensis* [9]. The classification controversy between morphological and molecular explorations mainly centers on the definition of *C. sumatraensis*. Specifically, in molecular studies, *C. milnedwardsii*, *C. thar*, and *C. maritimus*, which were previously classified as separate species based on morphology, are now included within the *C. sumatraensis* group [9]. Moreover, fossil evidence indicates that *C. sumatraensis* once existed in Northern China (Beijing) during the Pleistocene (about 2.58 Ma–11.7 ka), but the current range in China has been contracted to the southern area. Additionally, morphological investigation of *C. sumatraensis* indicated that their body size tended to increase from the mid-Pleistocene to the late Pleistocene, and then decreased in the Holocene [3].

At the molecular level, both the phylogeny and the classification of different serows have been investigated using mitochondrial sequences [9,10,11]. However, these molecular investigations are limited in extant specimens. The phylogenetic relationship between Quaternary *C. sumatraensis* and its extant counterparts remains poorly understood. Additionally, it is unclear whether the genetic diversity of *C. sumatraensis* was affected during the regional extirpation event that occurred at least in Northern China.

In this study, we aim to clarify the interspecific relationships of ancient and modern serows at the molecular level and therefore to explore their population history. We obtained two near-complete mitochondrial genomes (15,577 bp and 15,597 bp, both are out of 16,524 bp) from two ancient serow individuals and performed phylogenetic analyses of modern and ancient serows and updated the divergence time of *Capricornis*, as well as investigating the effective maternal population of *C. sumatraensis*. The new findings in this study provide further understanding into the evolutionary history of *Capricornis*.

## 2. Materials and Methods

### 2.1. Samples

Two ancient serow samples were collected from Beijing and Guizhou, China (Figure 1). One sample (CADG839) is stored at the Institute of Vertebrate Paleontology and Paleoanthropology. The other sample (CADG946) is stored at the China University of Geosciences (Wuhan). Two samples were sent for accelerator mass spectrometry (AMS) radiocarbon dating at the BETA laboratory, Miami, USA. The age of CADG839 is 9884–10,160 cal BP, and CADG946 is 2361–2703 cal BP (Appendix A).

### 2.2. Sample Handling and DNA Extraction

Prior to DNA extraction, the samples were cleaned with a brush and soaked in 0.5% bleach for 10 min, and then rinsed with nuclear-free water [13]. Each sample was ground into powder with a mortar and approximately 125 mg of powder was added into a 15 mL centrifuge tube that contained 3.0 mL EDTA (0.5 M, pH = 8) and 40 µL proteinase K (20 mg/mL). The extraction buffer with the sample powder was incubated overnight in a rotating hybridization oven at 37 °C. The incubation mix was centrifuged at 7000 rpm for 10 min, and the supernatant was transferred into an ultrafiltration tube (Millipore, Darmstadt, Germany) and condensed to about 100 µL by centrifuging at 7000 rpm for 39 min. Finally, DNA was purified and eluted in 50 µL EB using a MinElute PCR Purification Kit (Qiagen, Hilden, Germany), according to the manual instructions.

### 2.3. DNA Library Preparation and Sequencing

Each of 20 µL extractions was used for double-stranded Illumina libraries construction, modified from Meyer and Kircher [14]. In the blunt-end repair process, the following components were used: 5 µL NEB buffer 2 (New England Biolabs, Ipswich, UK), 5 µL ATP (New England Biolabs, Ipswich, UK), 2 µL BSA (New England Biolabs, Ipswich, UK), 2 µL T4 DNA polymerase (New England Biolabs, Ipswich, UK), 0.4 µL T4 polynucleotide kinase (New England Biolabs, Ipswich, UK), and 2 µL dNTPs (Tiangen, Beijing, China). The mixture was then supplemented with 20 µL DNA template and nuclease-free water to a final volume of 50 µL. The reaction was incubated at 15 °C for 15 min, followed by an additional incubation at 25 °C for 15 min using a PCR machine. Subsequently, the mixed reaction was purified using the MinElute PCR Purification Kit (Qiagen, Hilden, Germany), and the DNA was eluted in 21 µL of TET buffer. In the adapter ligation step, the following components were used: 1 µL Adapter (dilution 1:20) (Sangon Biotech, Shanghai, China), 1 µL Quick Ligase (New England Biolabs, Ipswich, UK), 20 µL Quick Ligase buffer (New England Biolabs, Ipswich, UK), and 18 µL of template. The reaction mixture was incubated at 22 °C for 15 min using a PCR machine. Subsequently, the mixed reaction was purified using the MinElute PCR purification kit (Qiagen, Hilden, Germany), and the DNA was eluted in 23 µL TET buffer. In addition, for the adaptor fill-in reaction, we used the following components: 4 µL isothermal buffer (New England Biolabs, Ipswich, UK), 2 µL Bst polymerase (New England Biolabs, Ipswich, UK), 2 µL dNTPs, 20 µL DNA template, and water to achieve a final reaction volume of 40 µL. The reaction mixture was incubated at 37 °C for 20 min using a PCR machine, followed by a temperature increase to 80 °C, and incubated for 20 min. The resulting reaction product was used for PCR amplification. For the indexing PCR amplifications, we prepared the reactions with a Q5 Hot Start High-Fidelity 2× Master Mix (New England Biolabs, Ipswich, UK) using the following cycling protocol: an initial denaturation step at 98 °C for 30 s, followed by 17 cycles of denaturation at 98 °C for 10 s, annealing at 60 °C for 75 s, and extension at 60 °C for 6 min. The library quality was measured using Qubit 4.0 (Invitrogen, Carlsbad, CA, USA) and TapeStation 4150 (Agilent, Santa Clara, CA, USA). To detect contamination, blank controls were set during the DNA extraction, library construction, and PCR amplification procedures. Finally, all libraries were sequenced on an Illumina NovaSeq6000 platform at Annoroad Gene Technology Co., Ltd., Beijing, China.

### 2.4. Data Analysis

The adapter sequence of raw reads was excised with fastp v 0.21.0 [15]. After we had compared the reads mapping to four *Capricornis* mitochondrial genomes (Appendix A), we finally mapped the trimmed reads to the *C. sumatraensis* mitochondrial genome (GenBank No. NC020629) using the “aln” algorithm in a Burrows–Wheeler aligner (BWA v 0.7.17) with default parameters [16]. Reads with a mapping quality below 30 were discarded using “view”, duplicate reads were removed using “rmdup”, and alignments were sorted using “sort” in SAMtools v 1.9 [17]. All BAM files of libraries were then merged, and duplications were removed to produce a combined BAM file using SAMtools v 1.9. Finally, consensus sequences were constructed using “doFasta 3” in ANGSD v. 0.921 [18]. Read coverage across the reference was calculated using Qualimap v 2.2.1 [19]. The base damage to DNA fragments was analyzed via mapDamage2 with default parameters [20]. Please refer to the Appendix A for detailed information on the sequencing quality of all libraries.

### 2.5. Bioinformatics Analysis

To investigate the phylogenetic relationship within the genus *Capricornis*, our two newly obtained near-complete mitochondrial sequences and 23 mitochondrial genomes retrieved from GenBank (18 available *Capricornis*, and 5 outgroup sequences that include 3 *Naemorhedus goral*, 1 *Ovibos moschatus*, 1 *Ovis aries*; see Appendix A) were aligned with MAFFT on the CIPRES portal and manually inspected after alignment [21,22]. Homologous sequences were obtained after we had removed the high variable control region (CR). Phylogenetic trees were constructed using neighbor-joining (NJ) and maximum-likelihood (ML) methods in MEGA 11 [23]. The NJ tree was constructed based on a Jukes–Cantor distance. The ML tree was constructed with the heuristic method of the nearest-neighbor-interchange, and the optimal substitution model ”GTR + I + G” was selected through comparison of Bayesian Information Criterion (BIC) scores in a jModelTest v2.1 [24]. A total of 1000 bootstrap replicates was used to ensure the reliability of nodes [25].

In order to reveal the interspecies relationship of serows, a median-joining network was reconstructed with Popart 1.7 using 2 near-complete mitochondrial DNA sequences in this study and 18 complete mitochondrial DNA sequences of *Capricornis* from the GenBank (Appendix A) [26]. Only sites covered by all individuals were retained to obtain a length of 14,034 bp dataset.

Bayesian analysis was conducted with the same dataset of the *Capricornis* phylogeny that includes 25 sequences, using the Monte Carlo Markov chains (MCMC) method implemented in BEAST 1.8.4 [27]. Due to the lack of direct fossil records for calibration, we used a comprehensive evolutionary framework of bovidae to estimate the divergence time between “*Capricornis*/*Naemorhedus*” and the most recent common ancestor (TMRCA) of caprini (Appendix A), drawing on the method of Bibi [28] and Pérez et al. [29]. Two node ages were used to correct our BEAST analysis: the divergence time between *Capricornis* and *Naemorhedus* with a mean of 4.91 Ma, and the TMRCA of caprini with a mean of 9.8 Ma. Meanwhile, the strict clock was set with reference to the mitochondrial substitution rate of caprini (substitution rate of 3.8–5.4 × 10^−8^ substitutions per nucleotide per year) [30], and a constant population size in parameter settings. MCMC ran for 75 million iterations, sampling every 1000 steps. Tracer v1.6 was used to test the effective sample size (ESS > 200), discarding the first 10% of samples as burn-in [31]. A maximum clade credibility (MCC) tree was visualized using TreeAnnotator v1.5.4 [27], and modified in FigTree v1.4.4 (http://tree.bio.ed.ac.uk/software/figtree, accessed on 1 May 2023). The Bayesian skyline plots (BSPs) of *C. sumatraensis* were simulated with identical parameters in BEAST v1.8.4 and visualized in Tracer v1.6.3.

## 3. Results

After adapter trimming and short read removal, a total number of 2102 and 2934 unique reads for CADG839 and CADG946 was successfully mapped onto a *C. sumatraensis* mitochondrial sequence (NC020629). Two near-complete mitochondrial genomes were obtained from two serow subfossils (15,577 bp for CADG839, 15,597 bp for CADG946) with mean depths of 8.13 and 7.98 folds, respectively. The length of the DNA fragments of our samples was between 35 and 70 bp, with significant C to T misincorporation at the 5′ end of the reads, which is consistent with the typical ancient DNA damage pattern (Appendix A) [32].

Both our Bayesian phylogenetic MCC tree (Figure 2) and the ML phylogenetic tree (Appendix A) indicate that there are four clades at the molecular level in *Capricornis* that correspond to four species, i.e., *C. crispus*, *C. swinhoei*, *C. rubidus*, and *C. sumatraensis*. Additionally, our analysis revealed that the ancient serows in this study are within the maternal diversity of modern ones, i.e., one of our two samples, CADG839, is settled in *C. sumatraensis* subclade A2 and close to a modern individual from Sichuan province, China; while the other sample, CADG946, is placed in the basal position of subclade A1 that mainly consists of samples from South China.

Regarding the divergence time of different serow clades, we estimated that the first divergence occurred in the *C. crispus* clade from the serow common ancestor around 2.37 Ma (95% HPD: 2.02–2.74 Ma), followed by the split of the *C. swinhoei* clade around 1.7 Ma (95% HPD: 1.5–1.96 Ma). The divergence time between the *C. rubidus* clade and the *C. sumatraensis* clade was approximately 0.7 Ma (95% HPD: 0.58–0.85 Ma), whereas differentiation within the *C. sumatraensis* clade began circa 0.3 Ma (95% HPD: 0.25–0.37 Ma) (Figure 2).

The result of haplogroup network analysis further supports the finding of phylogenetic analysis, indicating that the four major haplogroups correspond to the four species delineated at the molecular level, i.e., *C. crispus*, *C. swinhoei*, *C. rubidus*, and *C. sumatraensis*. Additionally, the *C. sumatraensis* haplogroup can be further subdivided into two secondary haplogroups (*C. sumatraensis* A and *C. sumatraensis B*) (Figure 3).

The Bayesian skyline plot analysis indicates that the effective maternal population of *C. sumatraensis* increased slightly from about 225 Kya to 160 Kya, followed by a stable stage until 90 Kya, and then went to another increase period until 50 Kya and remained stable in the most recent 50,000 years (Figure 4).

## 4. Discussion

### 4.1. Capricornis Phylogeny and Their Divergences

*Capricornis* have been morphologically classified into seven species, while only four species have been defined at the molecular level [6,7,8,9]. In this study, with the adding of the ancient *Capricornis* individuals, we observed four *Capricornis* mitochondrial clades that agree with previous molecular investigation of four species by Mori et al. [9] (Figure 2 and Appendix A). However, considering that different species or populations could be mixed in the mitochondrial phylogeny in other mammals, such as the African forest elephant and the straight-tusked elephant, the extinct Eurasian cave hyena and the African spotted hyena [33,34], we suspect that the exploration of nuclear genomes of different *Capricornis* groups may provide better phylogenetic distinctions between serows.

Geist (1987) suggested that *C. crispus* and *C. swinhoei* are likely the product of adaptive radiation of mainland serows [35]. It is possible that they were transported to Japan and Taiwan island through a medium such as a land bridge, and subsequently became geographically isolated on the islands [36,37,38]. A previous molecular study based on partial mitochondrial sequence has estimated that the first separation occurred between 0.67 Ma and 1.07 Ma, when *C. crispus* diverged from the ancestor group [10]. A recent study by Dou et al. utilizing complete mitochondrial genome estimated that *Capricornis* and *Naemorhedus* shared a common ancestor approximately 3.79 Ma, and the divergence times of *C. crispus* and *C. swinhoei* were estimated around 1.88 Ma and 1.27 Ma, respectively [11]. In this study, we estimated earlier time scales than those noted in Dou et al., with regard to both the *Capricornis* origin and its divergences. Specifically, we estimated that *Capricornis* and *Naemorhedus* had the most recent common ancestor approximately 4.83 Ma, and the divergence times of *C. crispus* and *C. swinhoei* were calculated approximately 2.41 Ma and 1.73 Ma, respectively (Figure 2). We suppose that the inconsistent results may be caused by different fossil calibration node selections. We used the updated fossil nodes of bovidae, i.e., 9.8 Ma for the TMRCA of caprini, and 4.91 Ma between *Capricornis* and *Naemorhedus*, which were considered to be exhibiting a smaller variance in estimated ages and thus to better reflect the fossil record [28], while Dou et al. chose a fossil calibration node of 4.1 Ma between *O. moschatus* and *C. sumatraensis* [11]. Furthermore, we found that the divergence times of these two species that we obtained are close to the times when Japan and Taiwan island became disconnected from the mainland. During the early Pleistocene, land bridges may have existed between the Eurasian continent and the Japanese archipelago due to geological and glacial activities, allowing terrestrial biota to migrate between these two areas [37]. It has been reported that Taiwan island became separated from the mainland in the late Pleistocene due to rising sea levels [36,38,39]. Therefore, we believe it is reasonable that the differentiation of *C. crispus* and *C. swinhoei* occurred after serows arrived in Japan and Taiwan island.

### 4.2. Phylogeography of C. sumatraensis

In both MCC and ML trees, there are two subclades within the *C. sumatraensis* clade (Figure 2 and Appendix A). Subclade A consists of the Sumatran serow from the Eurasian continent, while most individuals in subclade B are from Southwestern China and Northern Myanmar. Both of our two ancient samples (CADG839 and CADG946, dated 9884–10,160 cal BP, and 2361–2703 cal BP) cluster together with modern serow individuals in clade A, suggesting genetic continuity between Chinese ancient and modern serows. Moreover, fossil evidence indicates that *Capricornis* arrived in Northern China during the late Pleistocene, while the living range of the extant serows generally does not reach beyond 30° N (Figure 1) [1,3]. The phylogeny of CADG839, which is from Beijing, Northern China, is closely related to individuals from Sichuan, Southwestern China (MH155202) and Cambodia (NC020629), indicating that even the regional extirpation of the serow occurred in Northern China, the genetic diversity was not influenced, at least in the Holocene. Besides, we found that CADG946, the ancient individual from Guizhou, and all modern serow individuals from the same area are in different subclades. This means that there may be a regional replacement in Guizhou in terms of maternal lineage. Subclade A, which the ancient individual from Guizhou belongs to, contains both ancient and modern samples, ranging from Northern China to Cambodia, indicating that this subclade was once widely distributed on the mainland at the beginning of the Holocene.

It has been suggested that porcupine experienced local extirpation in Northern China in the mid-Holocene, while populations in the south survived, resulting in its contemporary distribution pattern [40]. Serow and porcupine shared similar habitats and experienced the same distribution changes according to fossil evidence. We therefore suppose that, similar to the strategy that porcupine adopted when it encountered frequent climate fluctuations, Holocene serows may make themselves adapt to environmental changes by moving southward from Northern China, accompanied by inner regional replacement afterwards.

### 4.3. Maternal Demographic History of the C. sumatraensis

Our BSP analysis reveals that the effective maternal population size of *C. sumatraensis* steadily increased from ~225 ka to ~160 ka and from ~90 ka to ~50 ka, and then remained stable for the past 50,000 years (Figure 4), indicating that there had been no reduction in its genetic diversity since MIS7, despite the occurrence of multiple glacial and interglacial cycles [41,42]. Moreover, the population growth in *C. sumatraensis* occurred in two cold periods (MIS6e–MIS6d and MIS5a–MIS4) [43,44], which may be linked to their adaptive ability. This may be due to their adaptive ability to different altitudes [45], such as migrating to lower altitude valleys during cold periods to alleviate survival pressures and gain stronger inter-species competitive advantages.

Notably, in contrast to the increased or stable mitochondrial genetic diversity indicated in our BSP analysis, recent studies suggest that both environmental changes and human activities have led to a decline in individual number in populations of *C. sumatraensis* [46,47,48]. We can infer that the decrease in individual numbers in its populations either has not affected its mitochondrial genetic diversity, or that there is a temporal latency to show its effect on genetic components. Thus far, it has been difficult to detect what the truth is due to limited molecular investigations of both ancient and modern *C. sumatraensis*. However, strengthening the protection of this vulnerable species, such as by reducing damage to its habitat and prohibiting illegal hunting, will help to alleviate the risk of turning it into an endangered species in both situations.

## 5. Conclusions

In summary, we presented the first molecular information on Holocene serows. We obtained four mitochondrial clades corresponding to four species initiated from a previous molecular investigation. We estimated earlier divergence times of different serows than previously suggested and detected connections between our estimation and the geographic isolation of Japan and Taiwan island. We confirmed that there is genetic continuity between Chinese ancient and modern *C. sumatraensis* individuals, although this species experienced regional extirpation in Northern China. Additionally, we found that there is no decrease in the maternal genetic diversity of the Sumatran serow even when the number of individuals in its population has declined under the pressure of environmental changes and human activities.

## Figures and Tables

**Figure 1 genes-14-01187-f001:**
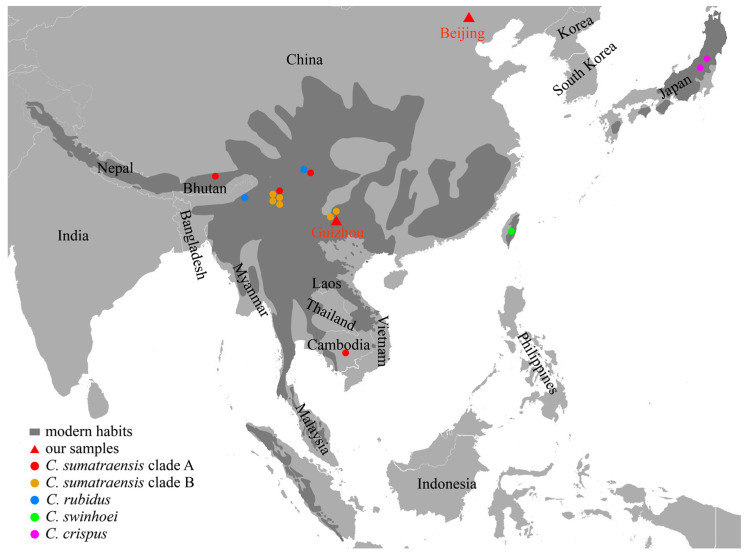
The geographical distribution of extant serows (dark grey) and locations of our two ancient samples (red triangles). Serow individuals with geographic information used in subsequent analyses are geographically marked with dots in different colors. Geographical distribution refers to the IUCN Red List [1,5,12]. We used the World Light Gray Base map provided by the ArcGIS platform as the basemap for our study area. The basemap is composed of a series of map tiles. The map tiles were used to provide a geographical reference for our study results, and we extracted a portion of Asia from it.

**Figure 2 genes-14-01187-f002:**
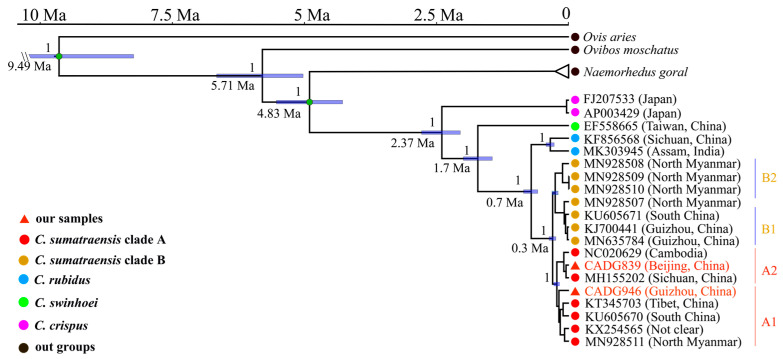
Mitochondrial phylogenetic relationships shown by MCC tree of 20 serows and 3 outgroup species. Two green dots indicate the nodes for calibration used in this study. The labels below the nodes indicate divergence age with blue bars showing 95% highest posterior density. Numbers above the nodes represent the Bayesan posterior probabilities. Our samples’ names are shown in red font. Location information is noted based on previous studies (see Appendix A).

**Figure 3 genes-14-01187-f003:**
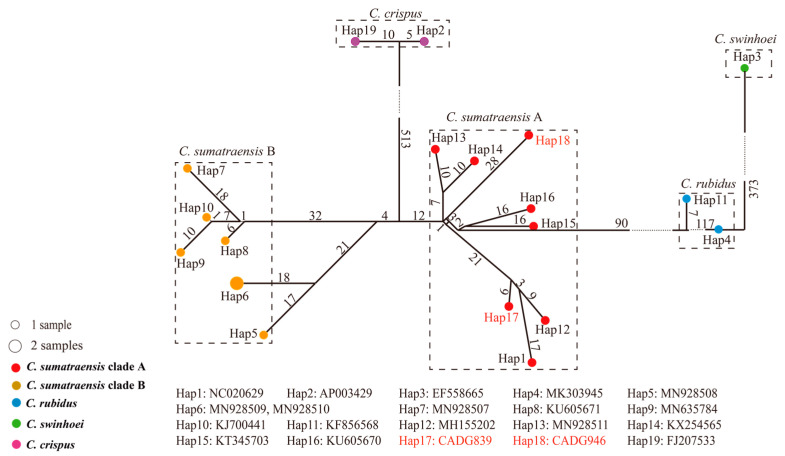
The haplogroup network analysis of serows. The number of samples is represented by the size of the circle. Different clades are represented by different colors. The numbers represent the count of mutation sites in the haplotype network, and our samples are shown in red font.

**Figure 4 genes-14-01187-f004:**
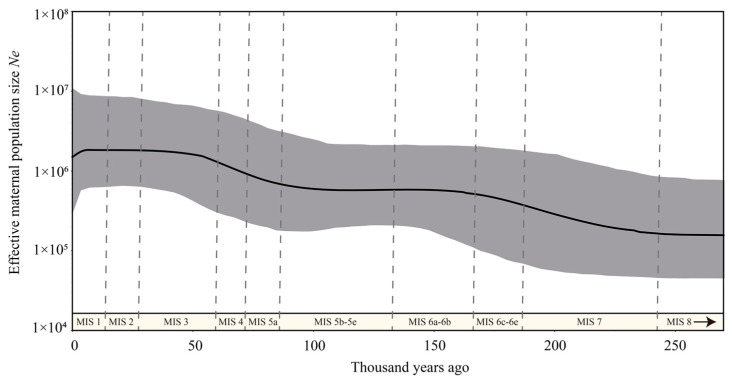
Bayesian skyline plots reconstructed for *C. sumatraensis.* The dataset used for BSP analysis consisted of all individuals within the *C. sumatraensis* (see Appendix A). Dotted lines show the boundaries of different marine isotope stages (MIS).

## Data Availability

Two ancient mitochondrial genomes have been submitted to the GenBank database under accession number OQ915127 (CADG839) and OQ915128 (CADG946). The data will be released on 28 April 2024. The address is as follows: GenBank www.ncbi.nlm.nih.gov/genbank, accessed on 28 April 2023.

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
