# Peer review of "Ancient Mitogenomes Reveal Stable Genetic Continuity of the Holocene Serows"

_genes, 2023, doi:10.3390/genes14061187_

Round 1

Reviewer 1 Report

Dear Authors,

well done for this work. However, I recommend that you go through it to improve the quality of the English so that some aspects would be made more clear to the reader.

Kindly find comments attached.

best wishes

The quality of english needs to be improved, even some ideas need to be written in a clearer manner.

Author Response

Dear reviewer,

Thanks for your appreciate on our work, as well as providing us very detailed and helpful suggestions on improvement of English writing in the PDF file. We accepted all your edits and comments in the revised version. We also went through the entire manuscript to improve the quality in terms of English writing. Our point-by-point response to reviewer 1 has been attached.

Sincerely,

Guilian Sheng

Reviewer 2 Report

I commend the authors for their very well-designed study. 

I just have some comments I will list here in addition to minor ones in the pdf attached

1- Why only mitochondrial DNA was tested here?

2- Why no target capture was used to enrich the mitochondrial or whole genome?

3- Where are the sequencing QC data, like how many libraries were sequenced for each sample, what was the total number of reads, dublication rates, etc?

Thank you

4- 

Author Response

Dear reviewer,

Thanks for your comments on our study. We have replied all your concerns via a point-by-point response as attached.

Sincerely,

Guilian Sheng

Round 2

Reviewer 1 Report

Dear authors,

well done for improving this manuscript. I am suggesting a few minor recommendations to finalize this work.

best regards

Dear authors,

I recommend that you go over the English again one final time as there were a few instances where some instances where the tense of the verbs was incorrect.